# Data-Driven Clustering via Parameterized Lloyd's Families

**Maria-Florina Balcan**
Department of Computer Science
Carnegie-Mellon University
Pittsburgh, PA 15213
ninamf@cs.cmu.edu

**Travis Dick**
Department of Computer Science
Carnegie-Mellon University
Pittsburgh, PA 15213
tdick@cs.cmu.edu

**Colin White**
Department of Computer Science
Carnegie-Mellon University
Pittsburgh, PA 15213
crwhite@cs.cmu.edu

## Abstract

Algorithms for clustering points in metric spaces is a long-studied area of research. Clustering has seen a multitude of work both theoretically, in understanding the approximation guarantees possible for many objective functions such as $k$-median and $k$-means clustering, and experimentally, in finding the fastest algorithms and seeding procedures for Lloyd's algorithm. The performance of a given clustering algorithm depends on the specific application at hand, and this may not be known up front. For example, a "typical instance" may vary depending on the application, and different clustering heuristics perform differently depending on the instance.

In this paper, we define an infinite family of algorithms generalizing Lloyd's algorithm, with one parameter controlling the initialization procedure, and another parameter controlling the local search procedure. This family of algorithms includes the celebrated $k$-means++ algorithm, as well as the classic farthest-first traversal algorithm. We design efficient learning algorithms which receive samples from an application-specific distribution over clustering instances and learn a near-optimal clustering algorithm from the class. We show the best parameters vary significantly across datasets such as MNIST, CIFAR, and mixtures of Gaussians. Our learned algorithms never perform worse than $k$-means++, and on some datasets we see significant improvements.

## 1 Introduction

Clustering is a fundamental problem in machine learning with applications in many areas including text analysis, transportation networks, social networks, and so on. The high-level goal of clustering is to divide a dataset into natural subgroups. For example, in text analysis we may want to divide documents based on topic, and in social networks we might want to find communities. A common approach to clustering is to set up an objective function and then approximately find the optimal solution according to the objective. There has been a wealth of both theoretical and empirical research in clustering using this approach Gonzalez [1985], Charikar et al. [1999], Arya et al. [2004], Arthur and Vassilvitskii [2007], Kaufman and Rousseeuw [2009], Ostrovsky et al. [2012], Byrka et al. [2015], Ahmadian et al. [2017].

The most popular method in practice for clustering is local search, where we start with $k$ centers and iteratively make incremental improvements until a local optimum is reached. For example, Lloyd's method (sometimes called $k$-means) Lloyd [1982] and $k$-medoids Friedman et al. [2001], Cohen et al. [2016] are two popular local search algorithms. There are multiple decisions an algorithm designer must make when using a local search algorithm. First, the algorithm designer must decide how to seed local search, e.g., how the algorithm chooses the $k$ initial centers. There is a large body of work on seeding algorithms, since the initial choice of centers can have a large effect on both the quality of the outputted clustering and the time it takes for the algorithm to converge Higgs et al. [1997], Pena et al. [1999], Arai and Barakbah [2007]. The best seeding method often depends on the specific application at hand. For example, a "typical problem instance" in one setting may have significantly different properties from that in another, causing some seeding methods to perform better than others. Second, the algorithm designer must decide on an objective function for the local search phase ($k$-means, $k$-median, etc.) For some applications, there is an obvious choice. For instance, if the application is Wi-Fi hotspot location, then the explicit goal is to minimize the $k$-center objective function. For many other applications such as clustering communities in a social network, the goal is to find clusters which are close to an unknown target clustering, and we may use an objective function for local search in the hopes that approximately minimizing the chosen objective will produce clusterings which are close to matching the target clustering (in terms of the number of misclassified points). As before, the best objective function for local search may depend on the specific application.

In this paper, we show positive theoretical and empirical results for learning the best initialization and local search procedures over a large family of algorithms. We take a transfer learning approach where we assume there is an unknown distribution over problem instances corresponding to our application, and the goal is to use experience from the early instances to perform well on the later instances. For example, if our application is clustering facilities in a city, we would look at a sample of cities with existing optimally-placed facilities, and use this information to find the empirically best seeding/local search pair from an infinite family, and we use this pair to cluster facilities in new cities.

$(\alpha, \beta)$**-Lloyds++** We define an infinite family of algorithms generalizing Lloyd's method, with two parameters $\alpha$ and $\beta$. Our algorithms have two phases, a seeding phase to find $k$ initial centers (parameterized by $\alpha$), and a local search phase which uses Lloyd's method to converge to a local optimum (parameterized by $\beta$). In the seeding phase, each point $v$ is sampled with probability proportional to $d_{\min}(v, C)^\alpha$, where $C$ is the set of centers chosen so far and $d_{\min}(v, C) = \min_{c \in C} d(v, c)$. Then Lloyd's method is used to converge to a local minima for the $\ell_\beta$ objective. By ranging $\alpha \in [0, \infty) \cup \{\infty\}$ and $\beta \in [1, \infty) \cup \{\infty\}$, we define our infinite family of algorithms which we call $(\alpha, \beta)$-Lloyds++. Setting $\alpha = \beta = 2$ corresponds to the $k$-means++ algorithm Arthur and Vassilvitskii [2007]. The seeding phase is a spectrum between random seeding ($\alpha = 0$), and farthest-first traversal Gonzalez [1985], Dasgupta and Long [2005] ($\alpha = \infty$), and the Lloyd's step is able to optimize over common objectives including $k$-median ($\beta = 1$), $k$-means ($\beta = 2$), and $k$-center ($\beta = \infty$). We design efficient learning algorithms which receive samples from an application-specific distribution over clustering instances and learn a near-optimal clustering algorithm from our family.

**Theoretical analysis** In Section 4, we prove that $O\left(\frac{1}{\epsilon^2} \min(T, k) \log n\right)$ samples are sufficient to guarantee the empirically optimal parameters $(\hat{\alpha}, \hat{\beta})$ have expected cost at most $\epsilon$ higher than the optimal parameters $(\alpha^*, \beta^*)$ over the distribution, with high probability over the random sample, where $n$ is the size of the clustering instances and $T$ is the maximum number of Lloyd's iterations. The key challenge is that for any clustering instance, the cost of the outputted clustering is not even a continuous function in $\alpha$ or $\beta$ since a slight tweak in the parameters may lead to a completely different run of the algorithm. We overcome this obstacle by showing a strong bound on the expected number of discontinuities of the cost function, which requires a delicate reasoning about the structure of the "decision points" in the execution of the algorithm; in other words, for a given clustering instance, we must reason about the total number of outcomes the algorithm can produce over the full range of parameters. This allows us to use Rademacher complexity, a distribution-specific technique for achieving uniform convergence.

Next, we complement our sample complexity result with a computational efficiency result. Specifically, we give a novel meta-algorithm which efficiently finds a near-optimal value $\hat{\alpha}$ with high probability. The high-level idea of our algorithm is to run depth-first-search over the "execution tree" of the algorithm, where a node in the tree represents a state of the algorithm, and edges represent a decision point. A key step in our meta-algorithm is to iteratively solve for the decision points of the algorithm, which itself is nontrivial since the equations governing the decision points do not have

a closed-form solution. We show the equations have a certain structure which allows us to binary search through the range of parameters to find the decision points.

**Experiments** We give a thorough experimental analysis of our family of algorithms by evaluating their performance on a number of different real-world and synthetic datasets including MNIST, Cifar10, CNAE-9, and mixtures of Gaussians. In each case, we create clustering instances by choosing subsets of the labels. For example, we look at an instance of MNIST with digits $\{0, 1, 2, 3, 4\}$, and also an instance with digits $\{5, 6, 7, 8, 9\}$. We show the the optimal parameters transfer from one instance to the other. Among datasets, there is no single parameter setting that is nearly optimal, and for some datasets, the best algorithm from the $(\alpha, \beta)$-Lloyds++ family performs significantly better than known algorithms such as $k$-means++ and farthest-first traversal.

## 2   Related Work

**Lloyd's method for clustering**   The iterative local search method for clustering, known as Lloyd's algorithm or sometimes called $k$-means, is one of the most popular algorithms for $k$-means clustering Lloyd [1982], and improvements are still being found Max [1960], MacQueen et al. [1967], Dempster et al. [1977], Pelleg and Moore [1999], Kanungo et al. [2002], Kaufman and Rousseeuw [2009]. Many different initialization approaches have been proposed Higgs et al. [1997], Pena et al. [1999], Arai and Barakbah [2007]. When using $d^2$-sampling to find the initial $k$ centers, the algorithm is known as $k$-means++, and the approximation guarantee is provably $O(\log k)$ Arthur and Vassilvitskii [2007].

**Learning to Learn**   A recent paper shows positive results for learning linkage-based algorithms with pruning over a distribution over clustering instances Balcan et al. [2017], although there is no empirical study done. There are several related models for learning the best representation and transfer learning for clustering. Ashtiani and Ben-David [2015] show how to learn an instance-specific embedding for a clustering instance, such that $k$-means does well over the embedding. There has also been work on related questions for transfer learning on unlabeled data and unsupervised tasks Raina et al. [2007], Yang et al. [2009], Jiang and Chung [2012]. To our knowledge, there is no prior work on learning the best clustering objective for a specific distribution over problem instances, given labeled clustering instances for training.

## 3   Preliminaries

**Clustering** A clustering instance $\mathcal{V}$ consists of a point set $V$ of size $n$, a distance metric $d$ (such as Euclidean distance in $\mathbb{R}^d$), and a desired number of clusters $1 \leq k \leq n$. A clustering $\mathcal{C} = \{C_1, \ldots, C_k\}$ is a $k$-partitioning of $V$. Often in practice, clustering is carried out by approximately minimizing an objective function (which maps each clustering to a nonzero value). Common objective functions such as $k$-median and $k$-means come from the $\ell_p$ family, where each cluster $C_i$ is assigned a center $c_i$ and $\text{cost}(\mathcal{C}) = \left( \sum_i \sum_{v \in C_i} d(v, c_i)^p \right)^{\frac{1}{p}}$ ($k$-median and $k$-means correspond to $p = 1$ and $p = 2$, respectively). There are two distinct goals for clustering depending on the application. For some applications such as computing facility locations, the algorithm designer's only goal is to find the best centers, and the actual partition $\{C_1, \ldots, C_k\}$ is not needed. For many other applications such as clustering documents by subject, clustering proteins by function, or discovering underlying communities in a social network, there exists an unknown "target" clustering $\mathcal{C}^* = \{C_1^*, \ldots, C_k^*\}$, and the goal is to output a clustering $\mathcal{C}$ which is close to $\mathcal{C}^*$. Formally, we define $\mathcal{C}$ and $\mathcal{C}'$ to be $\epsilon$-close if there exists a permutation $\sigma$ such that $\sum_{i=1}^{k} |C_i \setminus C'_{\sigma(i)}| \leq \epsilon n$. For these applications, the algorithm designer chooses an objective function while hoping that minimizing the objective function will lead to a clustering that is close to the target clustering. In this paper, we will focus on the cost function set to the distance to the target clustering, however, our analysis holds for an abstract cost function `cost` which can be set to an objective function or any other well-defined measure of cost.

**Algorithm Configuration** In this work, we assume that there exists an unknown, application-specific distribution $\mathcal{D}$ over a set of clustering instances such that for each instance $\mathcal{V}$, $|V| \leq n$. We suppose there is a cost function that measures the quality of a clustering of each instance. As discussed in the previous paragraph, we can set the cost function to be the expected Hamming distance of the returned clustering to the target clustering, the cost of an $\ell_p$ objective, or any other function. The

learner's goal is to find the parameters $\alpha$ and $\beta$ that approximately minimize the expected cost with respect to the distribution $\mathcal{D}$. Our main technical results bound the intrinsic complexity of the class of $(\alpha, \beta)$-Lloyds++ clustering algorithms, which leads to generalization guarantees through standard Rademacher complexity Bartlett and Mendelson [2002], Koltchinskii [2001]. This implies that the empirically optimal parameters are also nearly optimal in expectation.

# 4 $(\alpha, \beta)$-Lloyds++

In this section, we define an infinite family of algorithms generalizing Lloyd's algorithm, with one parameter controlling the the initialization procedure, and another parameter controlling the local search procedure. Our main results bound the intrinsic complexity of this family of algorithms (Theorems 4 and 5) and lead to sample complexity results guaranteeing the empirically optimal parameters over a sample are close to the optimal parameters over the unknown distribution. We measure optimality in terms of agreement with the target clustering. We also show theoretically that no parameters are optimal over all clustering applications (Theorem 2). Finally, we give an efficient algorithm for learning the best initialization parameter (Theorem 7).

Our family of algorithms is parameterized by choices of $\alpha \in [0, \infty) \cup \{\infty\}$ and $\beta \in [1, \infty) \cup \{\infty\}$. Each choice of $(\alpha, \beta)$ corresponds to one local search algorithm. A summary of the algorithm is as follows (see Algorithm 1). The algorithm has two phases. The goal of the first phase is to output $k$ initial centers. Each center is iteratively chosen by picking a point with probability proportional to the minimum distance to all centers picked so far, raised to the power of $\alpha$. The second phase is an iterative two step procedure similar to Lloyd's method, where the first step is to create a Voronoi partitioning of the points induced by the initial set of centers, and then a new set of centers is chosen by computing the $\ell_\beta$ mean of each Voronoi tile.

---
**Algorithm 1** $(\alpha, \beta)$-Lloyds++ Clustering

---
**Input:** Instance $\mathcal{V} = (V, d, k)$, parameter $\alpha$.
**Phase 1: Choosing initial centers with $d^\alpha$-sampling**
1. Initialize $C = \emptyset$ and draw a vector $\vec{Z} = \{z_1, \ldots, z_k\}$ from $[0, 1]^k$ uniformly at random.
2. For each $t = 1, \ldots, k$:
   (a) Partition $[0, 1]$ into $n$ intervals, where there is an interval $I_{v_i}$ for each $v_i$ with size equal to the probability of choosing $v_i$ during $d^\alpha$-sampling in round $t$ (see Figure 1).
   (b) Denote $c_t$ as the point such that $z_t \in I_{c_t}$, and add $c_t$ to $C$.
**Phase 2: Lloyd's algorithm**
5. Set $C' = \emptyset$. Let $\{C_1, \ldots, C_k\}$ denote the Voronoi tiling of $V$ induced by centers $C$.
6. Compute $\operatorname{argmin}_{x \in V} \sum_{v \in C_i} d(x, v)^\beta$ for all $1 \leq i \leq k$, and add it to $C'$.
7. If $C' \neq C$, set $C = C'$ and goto 5.
**Output:** Centers $C$ and clustering induced by $C$.

---

Our goal is to find parameters which return clusterings close to the ground-truth in expectation. Setting $\alpha = \beta = 2$ corresponds to the $k$-means++ algorithm. The seeding phase is a spectrum between random seeding ($\alpha = 0$), and farthest-first traversal ($\alpha = \infty$), and the Lloyd's algorithm can optimize for common clustering objectives including $k$-median ($\beta = 1$), $k$-means ($\beta = 2$), and $k$-center ($\beta = \infty$).

We start with two structural results about the family of $(\alpha, \beta)$-Lloyds++ clustering algorithms. The first shows that for sufficiently large $\alpha$, phase 1 of Algorithm 1 is equivalent to farthest-first traversal. This means that it is sufficient to consider $\alpha$ parameters in a bounded range.

Farthest-first traversal Gonzalez [1985] starts by choosing a random center, and then iteratively choosing the point farthest to all centers chosen so far, until there are $k$ centers. We assume that ties are broken uniformly at random.

**Lemma 1.** *Given a clustering instance $\mathcal{V}$ and $\delta > 0$, if $\alpha > \frac{\log\left(\frac{nk}{\delta}\right)}{\log s}$, then $d^\alpha$-sampling will give the same output as farthest-first traversal with probability $> 1 - \delta$. Here, $s$ denotes the minimum ratio $d_1/d_2$ between two distances $d_1 > d_2$ in the point set.*

For some datasets, $\frac{1}{\log s}$ might be very large. In Section 5, we empirically observe that for all datasets we tried, $(\alpha, \beta)$-Lloyds++ behaves the same as farthest-first traversal for $\alpha > 20$. Also, in the full version of this paper, we show that if the dataset satisfies a stability assumption called separability Kobren et al. [2017], Pruitt et al. [2011], then $(\alpha, \beta)$-Lloyds++ outputs the same clustering as farthest-first traversal with high probability when $\alpha > \log n$.

Next, to motivate learning the best parameters, we show that for *any* pair of parameters $(\alpha^*, \beta^*)$, there exists a clustering instance such that $(\alpha^*, \beta^*)$-Lloyds++ outperforms all other values of $\alpha, \beta$. This implies that $d^\beta$-sampling is not always the best choice of seeding for the $\ell_\beta$ objective. Let $\texttt{clus}_{\alpha,\beta}(\mathcal{V})$ denote the expected cost of the clustering outputted by $(\alpha, \beta)$-Lloyds++, with respect to the target clustering. Formally, $\texttt{clus}_{\alpha,\beta}(\mathcal{V}) = \mathbb{E}_{\vec{Z} \sim [0,1]^k} \left[ \texttt{clus}_{\alpha,\beta} \left( \mathcal{V}, \vec{Z} \right) \right]$, where $\texttt{clus}_{\alpha,\beta} \left( \mathcal{V}, \vec{Z} \right)$ is the cost of the clustering outputted by $(\alpha, \beta)$-Lloyds++ with randomness $\vec{Z}] \in [0,1]^k$ (see line 1 of Algorithm 1).

**Theorem 2.** *For $\alpha^* \in [0, \infty) \cup \{\infty\}$ and $\beta^* \in [1, \infty) \cup \{\infty\}$, there exists a clustering instance $\mathcal{V}$ whose target clustering is the optimal $\ell_{\beta^*}$ clustering, such that $\texttt{clus}_{\alpha^*,\beta^*}(\mathcal{V}) < \texttt{clus}_{\alpha,\beta}(\mathcal{V})$ for all $(\alpha, \beta) \neq (\alpha^*, \beta^*)$.*

**Sample efficiency** Now we give sample complexity bounds for learning the best algorithm from the class of $(\alpha, \beta)$-Lloyds++ algorithms. We analyze the phases of Algorithm 1 separately. For the first phase, our main structural result is to show that for a given clustering instance and value of $\beta$, with high probability over the randomness in Algorithm 1, the number of discontinuities of the cost function $\texttt{clus}_{\alpha,\beta} \left( \mathcal{V}, \vec{Z} \right)$ as we vary $\alpha \in [0, \alpha_h]$ is $O(nk(\log n)\alpha_h)$. Our analysis crucially harnesses the randomness in the algorithm to achieve this bound. For instance, if we use a combinatorial approach as in prior algorithm configuration work, we would only achieve a bound of $n^{O(k)}$, which is the total number of sets of $k$ centers. For completeness, we give a combinatorial proof of $O(n^{k+3})$ discontinuities in the full version of this paper.

To show the $O(nk(\log n)\alpha_h)$ upper bound, we start by giving a few definitions of concepts used in the proof. Assume we start to run Algorithm 1 without a specific setting of $\alpha$, but rather a range $[\alpha_\ell, \alpha_h]$, for some instance $\mathcal{V}$ and randomness $\vec{Z}$. In some round $t$, if Algorithm 1 would choose a center $c_t$ for every setting of $\alpha \in [\alpha_\ell, \alpha_h]$, then we continue normally. However, if the algorithm would choose a different center depending on the specific value of $\alpha$ used from the interval $[\alpha_\ell, \alpha_h]$, then we fork the algorithm, making one copy for each possible center. In particular, we partition $[\alpha_\ell, \alpha_h]$ into a finite number of sub-intervals such that the next center is constant on each interval. The boundaries between these intervals are "breakpoints", since as $\alpha$ crosses those values, the next center chosen by the algorithm changes. Our goal is to bound the total number of breakpoints over all $k$ rounds in phase 1 of Algorithm 1, which bounds the number of discontinuities of the cost of the outputted clustering as a function of $\alpha$ over $[\alpha_\ell, \alpha_h]$.

A crucial step in the above approach is determining when to fork and where the breakpoints are located. Recall that in round $t$ of Algorithm 1, each datapoint $v_i$ has an interval in $[0, 1]$ of size $\frac{d_i^\alpha}{D_n(\alpha)}$, where $d_i$ is the minimum distance from $v_i$ to the current set of centers, and $D_j(\alpha) = d_1^\alpha + \cdots + d_j^\alpha$. Furthermore, the interval is located between $\frac{D_{i-1}(\alpha)}{D_n(\alpha)}$ and $\frac{D_i(\alpha)}{D_n(\alpha)}$ (see Figure 1). WLOG, we assume $d_1 \geq \cdots \geq d_n$. We prove the following nice structure about these intervals.

**Lemma 3.** *Assume that $v_1, \ldots, v_n$ are sorted in decreasing distance from a set $C$ of centers. Then for each $i = 1, \ldots, n$, the function $\alpha \mapsto \frac{D_i(\alpha)}{D_n(\alpha)}$ is monotone increasing and continuous along $[0, \infty)$. Furthermore, for all $1 \leq i \leq j \leq n$ and $\alpha \in [0, \infty)$, we have $\frac{D_i(\alpha)}{D_n(\alpha)} \leq \frac{D_j(\alpha)}{D_n(\alpha)}$.*

This lemma guarantees two crucial properties. First, we know that for every (ordered) set $C$ of $t \leq k$ centers chosen by phase 1 of Algorithm 1 up to round $t$, there is a single interval (as opposed to a more complicated set) of $\alpha$-parameters that would give rise to $C$. Second, for an interval $[\alpha_\ell, \alpha_h]$, the set of possible next centers is exactly $v_{i_\ell}, v_{i_\ell+1}, \ldots, v_{i_h}$, where $i_\ell$ and $i_h$ are the centers sampled when $\alpha$ is $\alpha_\ell$ and $\alpha_h$, respectively (see Figure 1). Now we are ready to prove our main structural result. Formally, we define $\texttt{seed}_\alpha(\mathcal{V}, \vec{Z})$ as the outputted centers from phase 1 of Algorithm 1 on instance $\mathcal{V}$ with randomness $\vec{Z}$.

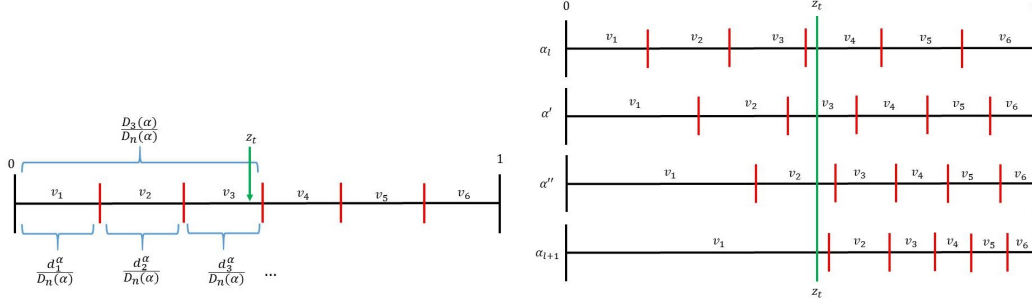

Figure 1: The algorithm chooses $v_3$ as a center (left). In the interval $[\alpha_\ell, \alpha_{\ell+1}]$, the algorithm may choose $v_4, v_3, v_2,$ or $v_1$ as a center, based on the value of $\alpha$ (right).

**Theorem 4.** *Given a clustering instance $\mathcal{V}$, the expected number of discontinuities of* $\mathtt{seed}_\alpha(\mathcal{V}, \vec{Z})$ *as a function of $\alpha$ over $[0, \alpha_h]$ is $O(nk(\log n)\alpha_h)$. Here, the expectation is over the uniformly random draw of $\vec{Z} \in [0, 1]^k$.*

*Proof sketch.* Consider round $t$ of a run of Algorithm 1. Suppose at the beginning of round $t$, there are $L$ possible states of the algorithm, e.g., $L$ sets of $\alpha$ such that within a set, the choice of the first $t-1$ centers is fixed. By Lemma 3, we can write these sets as $[\alpha_0, \alpha_1], \ldots, [\alpha_{L-1}, \alpha_L]$, where $0 = \alpha_0 < \cdots < \alpha_L = \alpha_h$. Given one interval, $[\alpha_\ell, \alpha_{\ell+1}]$, we claim the expected number of new breakpoints $\#I_{t,\ell}$ by choosing a center in round $t$ is bounded by $4n \log n(\alpha_{\ell+1} - \alpha_\ell)$. Note that $\#I_{t,\ell} + 1$ is the number of possible choices for the next center in round $t$ using $\alpha$ in $[\alpha_\ell, \alpha_{\ell+1}]$.

The claim gives an upper bound on the expected number of new breakpoints, where the expectation is only over $z_t$ (the uniformly random draw from $[0, 1]$ used by Algorithm 1 in round $t$), and the bound holds for any given configuration of $d_1 \geq \cdots \geq d_n$. Assuming the claim, we can finish off the proof by using linearity of expectation as follows. Let $\#I$ denote the total number of discontinuities of $\mathtt{seed}_\alpha(\mathcal{V}, \vec{Z})$.

$$E_{Z \in [0,1]^k}[\#I] \leq E_{Z \in [0,1]^k}\left[\sum_{t=1}^{k}\sum_{\ell=1}^{L}(\#I_{t,\ell})\right] \leq \sum_{t=1}^{k}\sum_{\ell=1}^{L} E_{Z \in [0,1]^k}[\#I_{t,\ell}] \leq 4nk \log n \cdot \alpha_h.$$

Now we will prove the claim. Given $z_t \in [0, 1]$, let $x$ and $y$ denote the minimum indices s.t. $\frac{D_x(\alpha_\ell)}{D_n(\alpha_\ell)} > z_t$ and $\frac{D_y(\alpha_{\ell+1})}{D_n(\alpha_{\ell+1})} > z_t$, respectively. Then from Lemma 3, the number of breakpoints is exactly $x - y$. Therefore, our goal is to compute $E_{z_t \in [0,1]}[x - y]$. One method is to sum up the expected number of breakpoints for each interval $I_v$ by bounding the maximum possible number of breakpoints given that $z_t$ lands in $I_v$. However, this will sometimes lead to a bound that is too coarse. For example, if $\alpha_{\ell+1} - \alpha_\ell = \epsilon \approx 0$, then for each bucket $I_{v_j}$, the maximum number of breakpoints is 1, but we want to show the expected number of breakpoints is proportional to $\epsilon$. To tighten up this analysis, we will show that for each bucket, the probability (over $z_t$) of achieving the maximum number of breakpoints is low.

Assuming that $z_t$ lands in a bucket $I_{v_j}$, we further break into cases as follows. Let $i$ denote the minimum index such that $\frac{D_i(\alpha_{\ell+1})}{D_n(\alpha_{\ell+1})} > \frac{D_j(\alpha_\ell)}{D_n(\alpha_\ell)}$. Note that $i$ is a function of $j, \alpha_\ell$, and $\alpha_{\ell+1}$, but it is independent of $z_t$. If $z_t$ is less than $\frac{D_i(\alpha_{\ell+1})}{D_n(\alpha_{\ell+1})}$, then we have the maximum number of breakpoints possible, since the algorithm chooses center $v_{i-1}$ when $\alpha = \alpha_{\ell+1}$ and it chooses center $v_j$ when $\alpha = \alpha_\ell$. The number of breakpoints is therefore $j - i + 1$, by Lemma 3. We denote this event by $E_{t,j}$, i.e., $E_{t,j}$ is the event that in round $t$, $z_t$ lands in $I_{v_j}$ and is less than $\frac{D_i(\alpha_{\ell+1})}{D_n(\alpha_{\ell+1})}$. If $z_t$ is instead greater than $\frac{D_i(\alpha_{\ell+1})}{D_n(\alpha_{\ell+1})}$, then the algorithm chooses center $v_i$ when $\alpha = \alpha_{\ell+1}$, so the number of breakpoints is $\leq j - i$. We denote this event by $E'_{t,j}$. Note that $E_{t,j}$ and $E'_{t,j}$ are disjoint and $E_{t,j} \cup E'_{t,j}$ is the event that $z_t \in I_{v_j}$.

Within an interval $I_{v_j}$, the expected number of breakpoints is

$$P(E_{t,j})(j - i + 1) + P(E'_{t,j})(j - i) = P(E_{t,j} \cup E_{t,j})(j - i) + P(E'_{t,j}).$$

We will show that $j - i$ and $P(E_{t,j})$ are both proportional to $(\log n)(\alpha_{\ell+1} - \alpha_\ell)$, which finishes off the claim.

First we upper bound $P(E_{t,j})$. Recall this is the probability that $z_t$ is in between $\frac{D_j(\alpha_\ell)}{D_n(\alpha_\ell)}$ and $\frac{D_i(\alpha_{\ell+1})}{D_n(\alpha_{\ell+1})}$, which we can show is at most $(4 \log n)(\alpha_{\ell+1} - \alpha_\ell)$ by upper bounding the derivative $\left| \frac{\partial}{\partial \alpha} \left( \frac{D_j(\alpha)}{D_n(\alpha)} \right) \right|$.

Now we upper bound $j - i$. Recall that $j - i$ represents the number of intervals between $\frac{D_i(\alpha_\ell)}{D_n(\alpha_\ell)}$ and $\frac{D_j(\alpha_\ell)}{D_n(\alpha_\ell)}$, and the smallest interval in this range is $\frac{d_j^{\alpha_\ell}}{D_n(\alpha_\ell)}$. Therefore, we can bound $j - i$ by dividing $\frac{D_j(\alpha_\ell)}{D_n(\alpha_\ell)} - \frac{D_i(\alpha_\ell)}{D_n(\alpha_\ell)}$ by $\frac{d_j^{\alpha_\ell}}{D_n(\alpha_\ell)}$. We can again use the derivative of $\frac{D_i(\alpha)}{D_n(\alpha)}$ to show this value is proportional to $4 \log n(\alpha_{\ell+1} - \alpha_\ell)$. This concludes the proof. $\square$

Now we analyze phase 2 of Algorithm 1. Since phase 2 does not have randomness, we use combinatorial techniques. We define $\texttt{lloyds}_\beta(\mathcal{V}, C, T)$ as the cost of the outputted clustering from phase 2 of Algorithm 1 on instance $\mathcal{V}$ with initial centers $C$, and a maximum of $T$ iterations.

**Theorem 5.** *Given $T \in \mathbb{N}$, a clustering instance $\mathcal{V}$, and a fixed set $C$ of initial centers, the number of discontinuities of $\texttt{lloyds}_\beta(\mathcal{V}, C, T)$ as a function of $\beta$ on instance $\mathcal{V}$ is $O(\min(n^{3T}, n^{k+3}))$.*

*Proof sketch.* Given $\mathcal{V}$ and a set of initial centers $C$, we bound the number of discontinuities introduced in the Lloyd's step of Algorithm 1. First, we give a bound of $n^{k+3}$ which holds for any value of $T$. Recall that Lloyd's algorithm is a two-step procedure, and note that the Voronoi partitioning step is independent of $\beta$. Let $\{C_1, \ldots, C_k\}$ denote the Voronoi partition of $V$ induced by $C$. Given one of these clusters $C_i$, the next center is computed by $\min_{c \in C_i} \sum_{v \in C_i} d(c, v)^\beta$. Given any $c_1, c_2 \in C_i$, the decision for whether $c_1$ is a better center than $c_2$ is governed by $\sum_{v \in C_i} d(c_1, v)^\beta < \sum_{v \in C_i} d(c_2, v)^\beta$. By a consequence of Rolle's theorem, this equation has at most $2n + 1$ roots. This equation depends on the set $C$ of centers, and the two points $c_1$ and $c_2$, therefore, there are $\binom{n}{k} \cdot \binom{n}{2}$ equations each with $2n + 1$ roots. We conclude that there are $n^{k+3}$ total intervals of $\beta$ such that the outcome of Lloyd's method is fixed.

Next we give a different analysis which bounds the number of discontinuities by $n^{3T}$, where $T$ is the maximum number of Lloyd's iterations. By the same analysis as the previous paragraph, if we only consider one round, then the total number of equations which govern the output of a Lloyd's iteration is $\binom{n}{2}$, since the set of centers $C$ is fixed. These equations have $2n + 1$ roots, so the total number of intervals in one round is $O(n^3)$. Therefore, over $T$ rounds, the number of intervals is $O(n^{3T})$. $\square$

By combining Theorem 4 with Theorem 5, and using standard learning theory results, we can bound the sample complexity needed to learn near-optimal parameters $\alpha, \beta$ for an unknown distribution $\mathcal{D}$ over clustering instances. Recall that $\texttt{clus}_{\alpha,\beta}(\mathcal{V})$ denotes the expected cost of the clustering outputted by $(\alpha, \beta)$-Lloyds++, with respect to the target clustering, and let $H$ denote the maximum value of $\texttt{clus}_{\alpha,\beta}(\mathcal{V})$.

**Theorem 6.** *Given $\alpha_h$ and a sample of size $m = O\left( \left( \frac{H}{\epsilon} \right)^2 \left( \min(T, k) \log n + \log \frac{1}{\delta} + \log \alpha_h \right) \right)$ from $\left( \mathcal{D} \times [0, 1]^k \right)^m$, with probability at least $1 - \delta$ over the choice of the sample, for all $\alpha \in [0, \alpha_h]$ and $\beta \in [1, \infty) \cup \{\infty\}$, $\left| \frac{1}{m} \sum_{i=1}^m \texttt{clus}_{\alpha,\beta} \left( V^{(i)}, \vec{Z}^{(i)} \right) - \mathbb{E}_{V \sim \mathcal{D}} [\texttt{clus}_{\alpha,\beta}(V)] \right| < \epsilon.$*

Note that a corollary of Theorem 6 and Lemma 1 is a uniform convergence bound for *all* $\alpha \in [0, \infty) \cup \{\infty\}$, however, the algorithm designer may decide to set $\alpha_h < \infty$.

**Computational efficiency** In this section, we present an algorithm for tuning $\alpha$ whose running time scales with the true number of discontinuities over the sample. Combined with Theorem 4, this gives a bound on the expected running time of tuning $\alpha$.

---

**Algorithm 2** Dynamic algorithm configuration

---

**Input:** Instance $\mathcal{V} = (V, d, k)$, randomness $\vec{Z}$, $\alpha_h$, $\epsilon > 0$
1. Initialize $Q$ to be an empty queue, then push the root node $(\langle\rangle, [0, \alpha_h])$ onto $Q$.
2. While $Q$ is non-empty
   (a) Pop node $(C, A)$ from $Q$ with centers $C$ and alpha interval $A$.
   (b) For each point $u_i$ that can be chosen as the next center, compute $A_i = \{\alpha \in A : u_i$ is the sampled center$\}$ up to error $\epsilon$ and set $C_i = C \cup \{u_i\}$.
   (c) For each $i$, if $|C_i| < k$, push $(C_i, A_i)$ onto $Q$. Otherwise, output $(C_i, A_i)$.

---

The high-level idea of our algorithm is to directly enumerate the set of centers that can possibly be output by $d^\alpha$-sampling for a given clustering instance $\mathcal{V}$ and pre-sampled randomness $\vec{Z}$. We know from the previous section how to count the number of new breakpoints at any given state in the algorithm, however, efficiently solving for the breakpoints poses a new challenge. From the previous section, we know the breakpoints in $\alpha$ occur when $\frac{D_i(\alpha)}{D_n(\alpha)} = z_t$. This is an exponential equation with $n$ terms, and there is no closed-form solution for $\alpha$. Although an arbitrary equation of this form may have up to $n$ solutions, our key observation is that if $d_1 \geq \cdots \geq d_n$, then $\frac{D_i(\alpha)}{D_n(\alpha)}$ must be monotone decreasing (from Lemma 3), therefore, it suffices to binary search over $\alpha$ to find the unique solution to this equation. We cannot find the exact value of the breakpoint from binary search (and even if there was a closed-form solution for the breakpoint, it might not be rational), however we can find the value to within additive error $\epsilon$ for all $\epsilon > 0$. We show that the expected cost function is $(Hnk \log n)$-Lipschitz in $\alpha$, therefore, it suffices to run $O\left(\log \frac{Hnk}{\epsilon}\right)$ rounds of binary search to find a solution whose expected cost is within $\epsilon$ of the optimal cost. This motivates Algorithm 2.

**Theorem 7.** *Given parameters $\alpha_h > 0$, $\epsilon > 0$, $\beta \geq 1$, and a sample $\mathcal{S}$ of size $m = O\left(\left(\frac{H}{\epsilon}\right)^2 \left(\min(T, k) \log n + \log \frac{1}{\delta} + \log \alpha_h\right)\right)$ from $\left(\mathcal{D} \times [0, 1]^k\right)^m$, run Algorithm 2 on each sample and collect all break-points (i.e., boundaries of the intervals $A_i$). With probability at least $1 - \delta$, the break-point $\bar{\alpha}$ with lowest empirical cost satisfies $|\text{clus}_{\bar{\alpha}, \beta}(\mathcal{S}) - \min_{0 < \alpha \leq \alpha_h} \text{clus}_{\alpha, \beta}(\mathcal{S})| < \epsilon$. The total running time to find the best break point is $O\left(mn^2 k^2 \alpha_h \log \left(\frac{nH}{\epsilon}\right) \log n\right)$.*

*Proof sketch.* First we argue that one of the break-points output by Algorithm 2 on the sample is approximately optimal. If we exactly solved for the break-points, then every value of $\alpha \in [0, \alpha_h]$ would produce exactly the same clusterings on the sample one of the break-points, so some break-point must be empirically optimal. By Theorem 6 this break-point is also approximately optimal in expectation. Since the algorithm approximately calculates the break-points to within additive error $\epsilon$, we are guaranteed that all true break-points are within distance $\epsilon$ of one output by the algorithm. In the full version of the paper, we show that the expected cost function is $(Hn^2 k \log n)$-Lipschitz. Therefore, the best approximate break-point has approximately optimal expected cost.

Now we analyze the runtime of Algorithm 2. Let $(C, A)$ be any node in the algorithm, with centers $C$ and alpha interval $A = [\alpha_\ell, \alpha_h]$. Sorting the points in $\mathcal{V}$ according to their distance to $C$ has complexity $O(n \log n)$. Finding the points sampled by $d^\alpha$-sampling with $\alpha$ set to $\alpha_\ell$ and $\alpha_h$ costs $O(n)$ time. Finally, computing the alpha interval $A_i$ for each child node of $(C, A)$ costs $O(n \log \frac{nH}{\epsilon})$ time, since we need to perform $\log \frac{nkH \log n}{\epsilon}$ iterations of binary search on $\alpha \mapsto \frac{D_i(\alpha)}{D_n(\alpha)}$ and each evaluation of the function costs $O(n)$ time. We charge this $O(n \log \frac{nH}{\epsilon})$ time to the corresponding child node. If we let $\#I$ denote the total number of $\alpha$-intervals for $\mathcal{V}$, then each layer of the execution tree has at most $\#I$ nodes, and the depth is $k$, giving a total running time of $O(\#I \cdot kn \log \frac{nH}{\epsilon})$. With Theorem 4 this gives us an expected runtime of $O\left(mn^2 k^2 \alpha_h (\log n) \left(\log \frac{nH}{\epsilon}\right)\right)$. $\square$

Since we showed that $d^\alpha$-sampling is Lipschitz as a function of $\alpha$, it is also possible to find the best $\alpha$ parameter with sub-optimality at most $\epsilon$ by finding the best point from a discretization of $[0, \alpha_h]$ with step-size $s = \epsilon/(Hn^2 k \log n)$. The running time of this algorithm is $O(n^3 k^2 H \log n/\epsilon)$, which is significantly slower than the efficient algorithm presented in this section. Intuitively, Algorithm 2 is able to binary search to find each breakpoint in time $O(\log \frac{nH}{\epsilon})$, whereas a discretization-based algorithm must check all values of alpha uniformly, so the runtime of the discretization-based algorithm increases by a multiplicative factor of $O\left(\frac{nH}{\epsilon} \cdot \left(\log \frac{nH}{\epsilon}\right)^{-1}\right)$.

# 5 Experiments

In this section, we empirically evaluate the effect of the $\alpha$ parameter on clustering cost for real-world and synthetic clustering domains. In the full version of this paper we provide full details and additional experiments exploring the effect of $\beta$ and the number of possible clusterings as we vary $\alpha$.

**Experiment Setup.** Our experiments evaluate the $(\alpha, \beta)$-Lloyds++ family of algorithms on distributions over clustering instances derived from multi-class classification datasets. For each classification dataset, we sample a clustering instance by choosing a random subset of $k$ labels, sampling $N$ examples belonging to each of the $k$ chosen labels. The clustering instance then consists of the $kN$ points, and the target clustering is given by the ground-truth labels. This sampling distribution covers many related clustering tasks (i.e., clustering different subsets of the same labels). We always measure distance between points using the $\ell_2$ distance and set $\beta = 2$. We measure clustering cost in terms of the *majority cost*, which is the fraction of points whose label disagrees with the majority label in their cluster. The majority cost takes values in $[0, 1]$ and is zero iff the output clustering matches the target clustering perfectly. We generate $m = 50,000$ samples from each distribution and divide them into equal-sized training and test sets. We then use Algorithm 2 to evaluate the average majority cost for all values of $\alpha$ on the train and test sets. Figure 2 shows the average majority cost for all values of $\alpha$ on both training and testing sets.

We ran experiments on datasets including MNIST, CIFAR10, CNAE9, and a synthetic Gaussian Grid dataset. For MNIST and CIFAR10 we set $k = 5$, and $N = 100$, while for CNAE9 and the Gaussian Grid we set $k = 4$ and $N = 120$. For MNIST, we used more samples ($m = 250,000$).

We find that the optimal value of $\alpha$ varies significantly between tasks, showing that tuning $\alpha$ on a per-task basis can lead to improved performance. Moreover, we find strong agreement in the average cost of each value of $\alpha$ across the independent training and testing samples of clustering instances, as predicted by our sample complexity results.

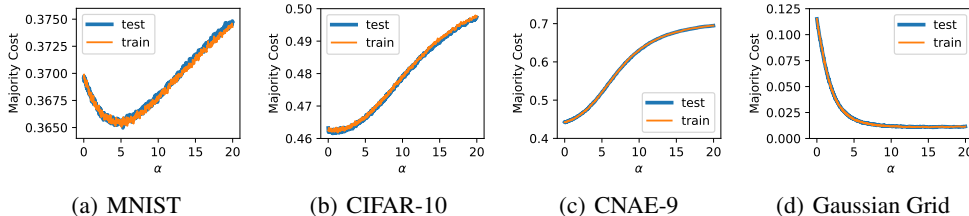

(a) MNIST      (b) CIFAR-10      (c) CNAE-9      (d) Gaussian Grid

Figure 2: Majority cost for $(\alpha, \beta)$-Lloyds++ as a function of $\alpha$ for $\beta = 2$.

# 6 Conclusion

We define an infinite family of algorithms generalizing Lloyd's method, with one parameter controlling the the initialization procedure, and another parameter controlling the local search procedure. This family of algorithms includes the celebrated $k$-means++ algorithm, as well as the classic farthest-first traversal algorithm. We provide a sample efficient and computationally efficient algorithm to learn a near-optimal parameter over an unknown distribution of clustering instances, by developing techniques to bound the expected number of discontinuities in the cost as a function of the parameter. We give a thorough empirical analysis, showing that the value of the optimal parameters transfer to related clustering instances. We show the optimal parameters vary among different types of datasets, and the optimal parameters often significantly improves the error compared to existing algorithms such as $k$-means++ and farthest-first traversal.

# 7 Acknowledgments

This work was supported in part by NSF grants CCF-1535967, IIS-1618714, an Amazon Research Award, a Microsoft Research Faculty Fellowship, a National Defense Science & Engineering Graduate (NDSEG) fellowship, and by the generosity of Eric and Wendy Schmidt by recommendation of the Schmidt Futures program.

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
