[Reviews · NeurIPS 2018]

Reviewer 1



The paper shows a family of data-driven clustering algorithms called (a,b)-loyds++. Parameter a has to do with the initialization phase, while parameter b has to do with the local search/optimization phase. One of the main technical and interesting results of the paper is the structural result that shows that for a given clustering instance, with high probability over the randomness in Algorithm 1, there are only polynomially many possible sets of output centers in phase 1 of the algorithm as parameter a varies. This is used for specifying the sample complexity as well as the computational complexity of the proposed algorithm. The experimental part of the paper is very short and laconic and it is very hard to infer the practical significance of the proposed method. In general, although many of the ideas in the paper are neat and they are nice combinations of existing ideas in learning theory, I am a bit skeptical about the practicality of the proposed algorithm.

Reviewer 2



The paper proposes a generalization of the KMeans++ Algorithm by introducing non-negative parameters alpha and beta. alpha parameterizes the initialization phase, i.e. the distribution from which the initial center seeds are sampled. beta parameterizes the objective according to which the data is clustered. The motivation is that different instances of clustering problems may cluster well according to different clustering objectives. The optimal parameter configuration of alpha and beta defines an optimal choice, from the proposed family of clustering algorithms. The paper offers several theoretical contributions. It provides guarantees for the number of samples necessary such that the empirically best parameter set yields clustering costs is within epsilon bounds of the optimal parameters. It provides an algorithm for the enumeration of all possible sets of initial centers for any alpha-interval. This allows to provide runtime guarantees for tuning the parameter alpha. The proposed algorithm is evaluated on MNIST, Cifar10, CNAE-9 and a mixture of Gaussians dataset, where the datasets are split into training and test sets. Results show that the optimal parameters transfer from one training to test set. The theoretical contribution of this paper seems solid. Proofs are provided for all theorems. The fact that a cost close to the empirically optimal value can be achieved for parameter beta = 2 for all datasets (see supplementary material), equaling to the k-means objective, is a bit disappointing. However, this finding might be different for different datasets. In general, the paper is difficult to read. This might be due to the fact that many details do not find place in the main paper, such that, without reading the proofs in detail, it becomes hard to understand the exact conditions under which the provided guarantees hold. For example on page 4, line 182, the paper need to be more precise! The assumption made is that ALL data points from one cluster have to be closer to ALL data points of the same cluster than to any point of a different cluster (which is a very strong assumption in practice).

Reviewer 3



This paper proposed an infinite family of algorithms that generalize Lloyd's algorithm with two controlling parameters. The optimal parameters can be learnt over an unknown distribution of clustering instances which is a data-driven approach. 1, The paper requires tuning parameters optimization. Since \alpha and \beta are correlated, can then be chosen simultaneously or iteratively instead of optimize one while fixing the other constant. Any guarantee the optimal values are global optimal? 2, What is the underlying true clusters are unbalanced/unevenly distributed? Is the initialization able to cover the minority clusters as well? It will be interesting to see how the optimal objective function look like. Since the loss function is always tricky in this situation. 3, The proposed method also requires a distance metric and a desired number of clusters (k) as inputs. The paper used l2 distance. Is the algorithm sensitive to the choice of distance measure and number of clusters? What happen if the predefined k is far away from the truth (eg. much smaller than truth)? Can those be included as model parameters and updated (data-driven) as well? 4, Any comparisons with other method in the empirical study?